# Treatment response in dogs with naturally occurring grade 3 elbow osteoarthritis following intra-articular injection of $^{117m}$Sn (tin) colloid

John Donecker[1]*, Michelle Fabiani[2], Lorrie Gaschen[3], Karanvir Singh Aulakh[3]

1 Exubrion Therapeutics, Buford, GA, United States of America, 2 Gulf Coast Veterinary Specialists, Houston, TX, United States of America, 3 School of Veterinary Medicine, Louisiana State University, Baton Rouge, LA, United States of America

* jdonecker@exubrion.com

**Data Availability Statement:** All supporting study data has been archived in Exubrion Therapeutics files including statistical analysis and is freely available with no restrictions upon request. Here is

## Abstract

The radionuclide $^{117m}$Sn (tin-117m) embedded in a homogeneous colloid is a novel radiosynoviorthesis (RSO) device for intra-articular (IA) administration to treat synovial inflammation and mitigate osteoarthritis (OA) in dogs. A study to evaluate tin-117m colloid treatment response in dogs with OA was conducted at two centers, the School of Veterinary Medicine at Louisiana State University, and at a referral practice in Houston, Texas. The tin-117m colloid was administered per-protocol to 14 client-owned dogs with radiographically confirmed, grade 3 OA in one or both elbow joints. Dog owners and attending clinicians assessed the level of pain at baseline (BL) and the post-treatment pain response at 90-day intervals for one year. Owners assessed treatment response according to a pain severity score (PSS) and a pain interference score (PIS) as defined by the Canine Brief Pain Inventory. Clinicians reported a lameness score using a 0–5 scale, from no lameness to continuous non-weight bearing lameness, when observing dogs at a walk and a trot. The rate of treatment success as determined by improved mean PSS and PIS scores reported by dog owners was >70% at all time points. Clinicians reported an improved mean pain score from BL at post-treatment Days 90 (p<0.05), 180, and 270. The dog owner and clinician assessments of treatment success were significantly correlated (p>0.05) at Day 90 and Day 180 time points. Results indicated that a single IA dose of tin-117m colloid provided a significant reduction in pain and lameness and improved functionality for up to a full year, with no adverse treatment related effects, in a high percentage of dogs with advanced, clinical OA of the elbow joint.

## Introduction

It has been reported that as many as 20% of dogs over 1 year of age are afflicted with osteoarthritis (OA) [1]. When it occurs in the canine elbow, OA most often develops as a sequela to elbow dysplasia [2], a common occurrence particularly in medium and large sized dogs [3, 4].

a publicly available link to the study report along with a listing of the appendices with relevant dataset titles: https://www.synovetin.com/resources/news.

**Funding:** Exubrion Therapeutics was the sole funding source for this study. Dr. John Donecker, Chief Veterinary Officer for Exubrion Therapeutics in association with Georgetown Clinical Consulting, Atlanta, GA designed the study at the request of Exubrion Therapeutics. Investigators at both centers adhered to the animal welfare standards as approved by the Louisiana State University Institutional Animal Care and Use Committee. This study was funded by amendment 2 of LSU protocol number 44181(KA) and the Veterinary Specialists of Texas, PC Clinical Site Agreement dated 6/15/2017 (MF). Owners of enrolled dogs signed consent forms indicating agreement for experimental RSO treatment and periodic reevaluations during the 1-year duration of the study. Exubrion Therapeutics was not involved with data collection or analysis however statistical review was conducted by Dr. Sheila Gross and funded by Exubrion Therapeutics. The decision to publish and prepare this manuscript was funded by Exubrion Therapeutics.

**Competing interests:** Drs. Aulakh and Fabiani are advisory board members for Exubrion Therapeutics and receive a small honorarium for consultation. Exubrion Therapeutics was the sole funding source for this study. We confirm that we adhere to all PLOS ONE policies on data and materials sharing.

The current approach to treating elbow OA is typically conservative and multimodal, consisting of a combination of weight management, non-steroidal anti-inflammatory drugs (NSAIDs), nutraceuticals, nutrition, intra-articular (IA) therapies and, less frequently, surgery [5]. Unfortunately, the OA pathology associated with elbow dysplasia is progressive and irreversible, and has a poor long-term prognosis. Treatment generally focuses on pain management and alleviation of clinical signs [4–7].

Histologic and imaging studies have confirmed that synovitis is the initial and dominant lesion in degenerative joint disease (DJD) and plays an important role in the pathophysiology of OA [8–10]. Catabolic and pro-inflammatory mediators produced by inflammation of the synovial membrane are associated with OA pathogenesis [1, 11, 12]. For example, elevated levels of local cytokines in human patients with early OA occur as a result of activation of the immune response in the synovial membrane [12]. The Multicenter Osteoarthritis Study (MOST) monitored 514 human subjects with magnetic resonance imaging (MRI)-detected effusion synovitis of the knee over 30 months. The investigators found that subjects had a 2.7-fold greater risk (p = 0.002) of cartilage loss compared with individuals without synovitis [13], a convincing indication of the chondro-destructive effects of synovitis.

It has been shown that early therapeutic intervention targeting the synovium can mitigate the cytokine cascade and alleviate clinical signs of joint disease, thus preventing, delaying, or limiting DJD progression [11, 12]. In fact, various conservative therapies currently used to treat OA, including NSAIDs and corticosteroids, are thought to work by reducing synovitis in affected joints [8, 14–16]. Radiosynoviorthesis (RSO) is a therapeutic approach to mitigating synovitis, where a low-energy ionizing radiation emitted by a radio-therapeutic radionuclide is used to penetrate the synovial membrane [17–20]. RSO has been successfully used in human medicine, particularly in Europe where it has been an accepted outpatient therapy for treatment of early-stage chronic synovitis in rheumatoid arthritis, psoriatric arthritis, hemophilic arthritis and OA patients for decades [19, 21]. The ionizing radiation results in macrophage apoptosis and ablation of inflammatory cytokines and growth factors within the synovium [19], reducing synovial hypertrophy, joint inflammation, and pain. Earlier studies of human and canine subjects diagnosed with chronic synovitis or OA revealed that radiation therapy is therapeutically effective and associated with favorable clinical outcomes over a 1-year period [2, 22, 23].

The radionuclide $^{117m}$Sn (tin-117m) embedded in a homogeneous colloid is a novel RSO preparation designed for IA administration to treat synovial inflammation and mitigate end-stage OA in cases of DJD in dogs [24]. Tin-117m emits conversion electrons, low-energy electrons released from the shell of the isotope as a result of radionuclide decay. Tin-117m conversion electrons have a short penetration range of approximately 300 μm (0.3 mm) in tissue and a half-life ($t_{1/2}$) of 14 days. These properties give tin-117m a precise range of activity that avoids exposing adjacent, non-target tissues to radiation. In addition, tin-117m has a duration of effect spanning several half-lives (approximately 42–90 days), which enables sustained therapeutic activity [25–27]. Although the energy activity declines relatively quickly, the post-treatment therapeutic effect lasts for up to a full year. No other radionuclide, or other treatment for canine OA pain and inflammation, has the radio-therapeutic properties of tin-117m.

The therapeutic goal of RSO is elimination of inflamed synoviocytes. Following IA injection, the tin-117m radionuclide particles are selectively phagocytized by synovial macrophages and synoviocytes and retained in situ [24]. A recent, well designed study in laboratory animals demonstrated that insufficient apoptosis of inflammatory macrophages in arthritic joints contributes to chronic progression of joint inflammation and destruction, characterized by super-abundance of osteoclasts and pro-inflammatory cytokines [10]. Based on the convincing outcomes of the study, the investigators concluded that mitigation of macrophage-dependent

synovial inflammation by means of macrophage apoptosis is a promising therapeutic strategy for treating inflammatory arthritis. The study is a conceptual affirmation of the RSO therapeutic pathway, including use of the tin-117m device described in this report.

While the colloid particles embedded with tin-117m are small enough (2–20 μm) to be phagocytized by synovial macrophages, they are too large to escape the confines of the joint [24]. Studies in laboratory animals have confirmed exceptionally high joint retention (>99%) of the homogeneous colloid containing tin-117m [26]. The energy emitted by tin-117m conversion electrons results in apoptosis of the activated synoviocytes and macrophages. A study in normal dogs confirmed that the limited penetration of tin-117m leaves adjacent non-synovial tissues unexposed, thus preserving the integrity and functionality of non-target tissues such as bone, cartilage, tendons, and ligaments [24].

A recent multi-center study found that dogs given a unilateral IA injection of tin-117m in elbow joints of dogs with grade 1 or 2 elbow OA had improved Canine Brief Pain Inventory (CBPI) scores and improved peak vertical force values from force-plate gait analysis at post-treatment intervals from 3 to 12 months, with no reported adverse effects [28]. As an extension of this earlier work in dogs with grade 1 or 2 OA lesions, t investigators at the same research centers evaluated the extent and duration of change in visual lameness and activity impairment as determined by dog owners and attending clinicians following unilateral or bilateral IA injections of tin-117m colloid in dogs with naturally occurring grade 3 elbow OA. Dog owner assessments utilized the CBPI instrument. Clinician assessments were based on a lameness score given following gait analysis of the dog while walking and trotting. The assessments were made at quarterly post-treatment intervals during a 1-year period. Rates of treatment success determined by the two assessment methods were then statistically correlated.

## Materials and methods

### Study animals

All dogs enrolled in the study (n = 15, 8 male, 7 female) were client owned pets residing with their caregivers. Dogs were at least 1 year of age (range 1.5–12 yrs, 11 mos), weighed ≥4.5 kg (range 19.8–45.7 kg), and had clinically evident, weight-bearing lameness in one or both elbow joints. Grade 3 elbow OA was radiographically confirmed in all dogs treated with IA tin-117m colloid. Dogs taking NSAIDs or other therapeutic medications for chronic treatment of joint pain were allowed to participate but had to be clinically lame while receiving NSAID treatment. Therapeutic medications were used by 13 of 15 dogs enrolled in the study, and are listed in Table 1. Dogs were excluded if they had any pain associated with shoulder or carpus joints based on orthopedic examination, or any comorbidity likely to preclude year-long participation in the study. Investigators at both centers adhered to the animal welfare standards as approved by the Louisiana State University Institutional Animal Care and Use Committee. Specifically, the LSU IACUC approved study protocol number 16–008 amendment 2 to allow the inclusion of dogs with grade 3 osteoarthritis in this study. In addition, owners of enrolled dogs signed consent forms indicating agreement for experimental RSO treatment and periodic reevaluations during the 1-year duration of the study.

### Diagnostic imaging

Study animals had radiographic and MRI confirmation of grade 3 OA in one or both elbows at the time of enrollment. A grade 3 elbow lesion was defined as well developed degenerative joint disease with new bone proliferation >5 mm along the proximal border of the anconeal process. Other aspects of the grade 3 lesion included indistinct or absent medial coronoid process with or without fragmentation, sclerosis of the semilunar notch, osteophytes on the cranial radial head,

**Table 1. Therapeutic medications used by study dogs at enrollment or during the study.**

| Medication[a] | No. dogs treated |
|---|---|
| Adequan | 3 |
| Antinol | 5 |
| Aspirin | 2 |
| Carprofen | 7 |
| Cosequin | 1 |
| Deracoxib | 1 |
| Firocoxib | 1 |
| Gabapentin | 3 |
| Glucosamine/chondroitin/MSM | 7 |
| Meloxicam | 2 |
| Movoflex soft chews | 1 |
| Omega 3 fatty acid/DHA | 6 |
| Phenobarbitol | 1 |
| Potassium bromide | 1 |
| Tramadol | 7 |

[a]Dosage size and frequency were variable.

periarticular osteophytes of the humeral condyle, and no evidence of ankylosis caused by periarticular new bone. Radiographs and MRI scans were obtained at baseline (BL) and at Day 365, with dogs positioned in sternal and lateral recumbency during the imaging procedure.

Each investigator followed a pre-determined quantitative morphometric protocol for MRI evaluation. Cranial and caudal joint-pouch width was measured on sagittal T2W sequences. The sagittal proton density (PD) was used to assess for the presence of hypointense synovial membrane thickness surrounded by hyperintense synovial fluid. A positive change was considered when an increase in fluid over time and a decrease in the synovial size occurred. The presence of synovial fluid heterogeneity that could be created by fibrin strands, was considered a negative change. Subchondral changes in bone intensity were assessed in sagittal and dorsal reformatted images using the 3D sequences optimized for each site. Sclerosis of the subchondral bone of the ulna at the base of the medial coronoid process and trochlear sulcus was assessed. Cartilage erosions were assessed by determining if the cartilage covered the joint uniformly and was assessed in the PD weighted sequences. The thickness of the joint capsule medial and lateral to the joint was measured on dorsal multiplanar reconstructed 3D scans. Images were evaluated by radiologist investigators using eFilm viewing software (IBM Watson Health, Armonk, NY, USA).

## Therapeutic device and administration procedure

The radio-therapeutic test article was a sterile, homogeneous colloid of tin-117m in ammonium hydroxide solution, supplied as 2–4 mCi (74–148 Mbq)/mL) suspension packaged in single patient-dose vials (Synovetin OA®, Exubrion Therapeutics, Buford, GA, USA). Each dose had a potency of 1.75 mCi for an IA dose for a 22.7 kg (50 lb) dog. Individual dosages were calculated based on the dog's body surface area adjusted relative to a 22.7 kg dog. The dosage was capped at 3.5 mCi per joint when the patient's weight exceeded 63.6 kg, with a total body dose ≤7.0 mCi in cases where a dog had both elbows treated.

Following sterile preparation of the injection site and using aseptic technique, the tin-117m colloid was administered by IA-articular injection with a 22-gauge needle while the dog was in

lateral recumbency during general anesthesia. Premedication for 12 of the enrolled dogs consisted of butorphanol (0.5 mg/kg intravenous) and dexmedetomidine (0.0005 mg/kg intramuscular). The two remaining dogs received butorphanol and dexmedetomidine at these dosages plus nalbuphine (0.5 mg/kg intramuscular) as the premedication regimen. After 15 min, anesthesia was induced with propofol (6 mg/kg intravenous) and maintained with inhaled isoflurane in oxygen. The procedure was performed using either a lateral or medial approach, depending on the investigator's preference. For the lateral approach, with the elbow flexed at 90 degrees, a spinal needle was inserted caudo-laterally between the lateral epicondylar crest of the humerus and the triceps tendon, then directed distal and slightly medial along the craniolateral aspect of the ulnar anconeus into the supratrochlear foramen of the humerus. For a medial approach, the needle was inserted approximately 1 cm distal to the medial epicondyle and directed perpendicular into the joint. After sterile, IA needle placement, 0.5 to 1.0 mL of joint fluid was removed for joint fluid analysis. The sterile tin-117m colloid was then injected slowly into the joint, followed by approximately 0.3 mL of air to minimize residual colloid in the syringe. Following needle removal, direct pressure was applied to the injection site for 2 minutes.

## Pain assessment methods

The CBPI, a survey instrument validated for canine OA and cancer [29–32], was used by dog owners to subjectively score their pet's pain severity and its functional impact on activity before and after treatment. Owners completed the 10-item survey at BL and on post-treatment Days 90, 180, 270, and 365. The mean value of CBPI items 1–4 represented a pain severity score (PSS), and the mean value of CBPI items 5–10 represented a pain interference score (PIS). The ten CBPI assessment items are listed in Table 2.

Attending veterinarians assessed lameness as determined by gait analysis in all dogs at BL and on Days 90, 180, 270, and 365. Dogs were led by a trained handler in a consistent manner and evaluated by a clinician observer. Lameness was separately determined during a walk and a trot and scored on a scale from 0 to 5 using the following grade definitions:

Grade 0 = No lameness, walks normally.

Grade 1 = Slight lameness.

Grade 2 = Obvious weight-bearing lameness.

Grade 3 = Severe weight-bearing lameness.

Grade 4 = Intermittent non-weight-bearing lameness.

Grade 5 = Continuous non-weight-bearing lameness

Although not part of the study protocol, three dogs were evaluated for lameness on an ad hoc basis by attending veterinarians using force-plate analysis.

## Study sites

The study was conducted at two centers, the School of Veterinary Medicine at Louisiana State University, and Gulf Coast Veterinary Specialists, a referral practice in Houston, Texas. Each site had a primary investigator who performed all diagnostic evaluations, clinician assessments of pain, and was responsible for compliance with the study protocol.

## Study design

Following diagnostic confirmation of single or bilateral grade 3 elbow OA, investigators calculated the appropriate tin-117m dosage for each participating dog, and the therapeutic device was administered. Dogs that were given the intended dose were classified as per-protocol (PP) animals. Dogs that received an injected dose >20% above or below the

**Table 2. Canine Brief Pain Inventory (CBPI) scores in dogs with grade 3 elbow osteoarthritis following per-protocol treatment with 117-tin colloid.**

| CBPI Item | Median Value (Range) at Study Interval | | | | |
|---|---|---|---|---|---|
| | Baseline | Day 90 | Day 180 | Day 270 | Day 365 |
| 1. Worst pain last 7 days[a] | 7.6 (5–9) | 4.5 (0–9) | 5.5 (0–8) | 3 (0–8) | 5 (0–8) |
| 2. Least pain last 7 days[a] | 4.5 (1–8) | 2.5 (0–7) | 2.5 (0–8) | 1 (0–6) | 2 (0–6) |
| 3. Ave. pain last 7 days[a] | 6 (2–8) | 3.5 (0–8) | 3.5 (0–7) | 1.5 (0–7) | 3 (0–7) |
| 4. Current pain[a] | 6 (0–8) | 3 (0–8) | 2.5 (0–7) | 1 (0–7) | 3 (0–8) |
| 5. General activity[b] | 7 (2–10) | 3.5 (0–8) | 4 (0–9) | 1.5 (0–6) | 3 (0–8) |
| 6. Enjoyment of life[b] | 6.5 (2–9) | 2.5 (0–8) | 4 (0–9) | 1 (0–7) | 3 (0–7) |
| 7. Ability to rise from laying down[b] | 7 (0–9) | 3 (0–9) | 5 (0–8) | 1.5 (0–5) | 3 (1–8) |
| 8. Ability to walk[b] | 6 (0–9) | 2 (0–8) | 3.5 (0–9) | 2 (0–7) | 2 (0–7) |
| 9. Ability to run[b] | 8 (1–10) | 2.5 (0–9) | 5.5 (0–10) | (0–7) | 3 (0–9) |
| 10. Ability to climb[b] | 7.5 (1–10) | 4.5 (0–9) | 3.5 (0–10) | 2 (0–7) | 2 (0–9) |
| **Mean PSS (items 1–4)** | **5.82 (2.0–8.0)** | **3.46\* (0–7.25)** | **3.54 (0–6.50)** | **2.33\* (0–7.0)** | **3.25\* (0–6.25)** |
| Mean PSS ≥1 improvement from BL | | 10/14 (71.4%) | 8/12 (66.7%) | 7/12 (70.0%) | 8/11 (72.7%) |
| **Mean PIS (items 5–10)** | **6.42 (1.17–8.83)** | **3.48 (0.17–8.33)** | **4.04 (0.17–7.17)** | **2.20\*\* (0–.650)** | **3.14 (0.17–8.00)** |
| Mean PIS ≥2 improvement from BL | | 8/14 (57.1%) | 7/12 (58.3%) | 9/10 (90.0%) | 7/11 (63.6%) |
| **Treatment success[c]** | | **10/14 (71.4%)** | **9/12 (75.0%)** | **9/10 (90.0%)** | **9/11 (81.82%)** |

BL = baseline, PIS = pain improvement score, PSS = pain severity score.

[a]Score assigned on scale of 0 (no pain) to 10 (extreme pain).

[b]Score associated with how pain has interfered with a specific activity, on scale of 0 (no interference) to 10 (completely interferes).

[c]Success = PSS reduced by ≥1.0 *or* PIS reduced by ≥2.0, i.e., individual dogs may have had treatment success as determined by PSS but not PIS, and vice-versa.

\*Average improvement significantly ($p < 0.05$) greater than 1.0 units compared to BL.

\*\*Average improvement significantly ($p < 0.05$) greater than 2.0 units compared to BL.

intended dose were designated as intent-to-treat (ITT) animals. Of the 15 enrolled dogs, 14 were designated as PP. The dog that was excluded from the PP population received 70.1% of the indicated tin-117m dose, a difference of >20%, and was assigned to the ITT group along with the 14 PP dogs for safety evaluation. Dogs were presented at the study sites for initial diagnosis, administration of the therapeutic agent, physical exams, and pain assessments performed at BL and Days 90, 180, 270, and 365. Dogs resided in their home settings between visits to the study sites.

Dog owner assessments of pain using the CBPI, and clinician assessments of pain using the walk-trot gait analysis were performed concurrently at BL and at four subsequent 90-day intervals for the one-year duration of the study. Treatment success determined by CBPI scoring was defined as improvement when the PSS score was reduced by ≥1.0 or when the PIS score was reduced by ≥2.0 when compared to BL. Treatment success determined by clinicians was defined by lameness scores that either declined or stayed the same compared to BL. The statistical correlation in treatment success rates when compared to BL for two assessment methods (CBPI and clinician assessments) was determined at each of the five study time points.

For purposes of evaluating product safety, clinical chemistry testing, CBC, urinalysis, and joint fluid analysis were performed for diagnostic samples obtained at BL and at 90-day post-treatment intervals. Joint fluid aspiration of the treated joint was performed for all dogs to evaluate IA cellular and protein composition. All testing of diagnostic samples was performed by an independent laboratory. Nuclear scintigraphy was performed for all treated joints to verify in situ presence of tin-117m at BL and Day 90.

## Statistical analysis

Data analyses were performed with statistical analysis software (SAS, version 9.4, SAS Institute Inc, Cary, NC). For those variables that were assessed for each elbow, the elbow was the experimental unit. For those variables that could only be evaluated for the whole animal, the dog was the experimental unit. For determination of treatment success, tests of statistical significance were completed at a two-sided alpha level of 0.05. For the CBPI and clinician assessments, comparisons were made for mean values obtained at BL and Days 90, 180, 270 and 365. The change in mean numerical values from one test interval to the next were also compared. Within group P-values were generated by the paired t-test or Wilcoxon signed rank test, depending on the distribution of the data, for clinical lameness assessments.

The data for PSS and PIS were analyzed using randomized-block Analysis of Variance, with time as a fixed effect and case as a random blocking effect. The baseline value was considered for use as a covariate, but was dropped from the model because after fitting the blocking factor there was no additional variability due to initial (baseline) differences. Post-treatment times were compared using a set of single-degree-of-freedom contrasts: Day 90 was compared to all later time points (Days 180, 270 ad 365); Day 180 was compared to the later time points (Days 270 and 365); and Day 270 was compared to Day 365. This partitioning of the degrees of freedom assumes that the responses over time are monotonic, i.e., that successive time points are either equal or all change in the same direction. The CBPI success criteria data were compared with the clinical lameness assessments of walk and trot for each dog. Two-by-two tables for treatment success were constructed for each visit. McNemar's test of agreement was applied to each metric. P-values ≥0.05 indicate agreement. Least-squares means were calculated for each time point.

# Results

## Client assessment of treatment response

Client assessments of pain severity and response to treatment of 14 dogs in the PP population are shown in Table 1. Both elbows were treated in12 dogs with grade 3 elbow OA. The other two dogs were treated in only one elbow as the contralateral elbow was a grade 0 normal elbow. The mean PSS score (items 1–4 in the CBPI) declined at all post-treatment time points by >2.0 points from 5.82 at BL. The mean PSS score of 3.46 at Day 90 and 2.33 at Day 270 were statistically significant improvements (p<0.05) in pain severity compared to the mean BL score. Similarly, the mean PIS score (items 5–10 in the CBPI) declined at all post-treatment time points by >2.0 points from 6.42 at BL. The mean PIS score of 3.48 at Day 90 and 2.20 at Day 270 were statistically significant (p<0.05) reductions in activity impairment compared to BL.

All ten items in the BL CPBI showed median pain scores of at least 4.5 (item 2, least pain in last 7 days), with median scores of some items as high as 7.6 (item 1, worst pain in last 7 days) and 8 (item 9, ability to run). The BL scores indicate that the study population was affected by marked levels of pain and pain-associated activity interference prior to treatment. Thus, the behavior of dogs in the PP population as indicated by the CBPI was consistent with the confirmed radiographic and MRI diagnosis of grade 3 elbow OA at BL.

The improvement of >2.0 in both the mean PSS and PIS scores met or exceeded the definition of treatment success for both pain assessment categories and was sustained at all time points for the 1-year duration of the study. The reduction from BL in the mean PSS score was particularly notable on Day 270, >3.0 points, as was the reduction in the mean PIS score on Days 270 and 365, >4.0 and >3.0 points, respectively.

Compared to BL, the rate of overall treatment success as determined by the CBPI was >70% at all time points. It should be noted that individual dogs may have met the criteria for treatment success as determined by PSS but not PIS, and vice-versa. Thus, the combined number of individuals successfully treated in either group could exceed the number of dogs successfully treated within the separate PSS or PIS groups. This occurred on Day 180 (9/12 dogs successfully treated) and Day 365 (9/11 dogs successfully treated).

## Clinician assessment of treatment response

The assessment of treatment response by clinicians based on walk and trot gait analysis is summarized in Table 3. The mean lameness score for both walk and trot was improved from BL at the Day 90 and 270 time points, and showed a non-significant increase at Day 365. The difference from BL in mean lameness scores for both gaits was significant (p<0.05) at Day 90. At Days 180, 270, and 365, the difference in mean lameness scores at walk and trot showed a gradual but non-significant increase from the previous visit.

## Agreement between client and clinician assessments

Treatment success rates for the dog owner and clinician assessments are compared in Table 4. For both the walk and trot gait analyses, clinician assessments of the treatment success rate

**Table 3. Clinician assessment of lameness score at walk or trot following treatment with 117-tin colloid.**

| Parameter | Time Point and Interval (No. Dogs) | Mean Lameness Score ± SEM | P-value |
|---|---|---|---|
| Lameness at walk | BL (n = 14) | 2.14 ± 0.18 | |
| | Day 90 (n = 14) | 1.29 ± 0.30 | |
| | Change from BL (n = 14) | | 0.01 |
| | Day 180 (n = 12) | 1.58 ± 0.36 | |
| | Change from BL (n = 12) | − 0.56 ± 0.34 | 0.17 |
| | Change from Last TP (n = 12) | 0.33 ± 0.19 | 0.10 |
| | Day 270 (n = 10) | 1.60 ± 0.34 | |
| | Change from BL (n = 10) | − 0.54 ± 0.34 | 0.27 |
| | Change from last TP (n = 10) | 0.30 ± 0.21 | 0.19 |
| | Day 365 (n = 11) | 2.18 ± 0.26 | |
| | Change from BL (n = 11) | 0.04 ± 0.30 | 0.55 |
| | Change from last TP (n = 10) | 0.60 ± 0.34 | 0.11 |
| Lameness at trot | BL (n = 13) | 2.15 ± 0.15 | |
| | Day 90 (n = 13) | 1.31 ± 0.31 | |
| | Change from BL (n = 13) | − 0.84 ± 0.25 | 0.01 |
| | Day 180 (n = 11) | 1.91 ± 0.28 | |
| | Change from BL (n = 11) | − 0.24 ± 0.18 | 0.34 |
| | Change from last TP (n = 11) | 0.73 ± 0.24 | 0.01 |
| | Day 270 (n = 10) | 1.90 ± 0.35 | |
| | Change from BL (n = 10) | − 0.25 ± 0.33 | 0.55 |
| | Change from last TP (n = 10) | 0.10 ± 0.23 | 1.000 |
| | Day 365 (n = 11) | 2.36 ± 0.24 | |
| | Change from BL (n = 11) | 0.21 ± 0.19 | 0.19 |
| | Change from last TP (n = 10) | 0.50 ± 0.27 | 0.10 |

BL = baseline, TP = time point.

**Table 4. Comparison of Canine Brief Pain Inventory (CBPI) assessment by dog owners and clinician assessment of treatment success following administration of tin-117m colloid.**

| Test Interval | CBPI Success Rate[a] | Clinician Assessment of Treatment Success | | | |
|---|---|---|---|---|---|
| | | Lameness at Walk | | Lameness at Trot | |
| | | Success Rate | P-Value[b] | Success Rate | P-Value[b] |
| BL to Day 90 | 10/14 (71.4%) | 8/14 (57.14%) | 0.6547 | 7/13 (59.85%) | 0.65 |
| BL to Day 180 | 9/12 (75.0%) | 4/12 (33.3%) | 0.4795 | 2/11 (18.18%) | 0.06 |
| BL to Day 270 | 9/10 (90.0%) | 5/10 (50.00%) | 0.0455 | 4/10 (40.00%) | 0.03 |
| BL to Day 365 | 9/11 (81.8%) | 3/11 (27.27%) | 0.0143 | 1/11 (9.09%) | 0.01 |

BL = baseline.

[a]Client assessments were for 14 dogs in the per-protocol treatment group.

[b]P-values ≥0.05 indicate agreement between CBPI and clinician assessments of success.

were in statistical agreement (p≥0.05) with the CPBI treatment success rates from BL to Day 90 and Day 180. The strongest correlation (p>0.06) between the two assessment methods was the interval from BL to Day 90. The difference in success rates for the CBPI and clinician assessments was not significant from BL to Days 270 and 365.

Fig 1 shows radiographs of grade 3 elbow OA lesions in one of the study dogs (no. 2-15L3). The radiographic progression of the culprit lesions observed at Day 365 was either static or mild. Table 5 compares the BL and post-treatment assessment scores for dog 2-15L3 given by the owner and clinician. The dog had CBPI scores of 5.25 and 1.17 for PSS and PIS, respectively, at BL, indicating observable pre-treatment levels of pain. Similarly, the attending clinician gave the patient a BL lameness score of 2.0, indicating obvious weight-bearing lameness. The dog's PSS improved by 4 to 5 points, and its PIS improved by 1.0 point following treatment. The clinician's gait analysis showed that the dog's lameness score at walk or trot improved by 1.0 point from BL to Day 90, and returned to its BL score at Day 365, an outcome supported by of the minimal extent of radiographic changes from BL to Day 365. This patient was one of three dogs in the study that was evaluated by force plate analysis. The dog in this case applied significantly greater peak vertical force on the affected limb at post-treatment Days 90, an objective indicator of intra-study reduction in lameness. Using the criteria previously described, both the owner and clinician considered dog 2-15L3 to be a treatment success that was sustained for the duration of the study. The case is useful as an example of what was often observed in the study population, namely unambiguous post-treatment improvement reported by dog owners, and a more conservative but still favorable assessment by clinicians, corroborated by a radiographic comparison of the affected joint at BL and Day 365.

## Safety evaluation

Clinical chemistry, CBC, and urinalysis values remained within normal ranges for all dogs for up to 12 months following treatment. Joint fluid analysis revealed no treatment related or clinically significant abnormalities. Scintigraphy results indicated that the tin-117m radionuclide was retained within the elbow joint of all dogs following IA injection and was still present at 90 days post-treatment.

## Discussion

All dogs enrolled in the study had severe, grade 3 elbow OA lesions, in most cases bilateral. Thus, the study evaluated tin-117m colloid treatment effectiveness in animals with advanced DJD, a considerable therapeutic challenge. Independent assessments of treatment response in

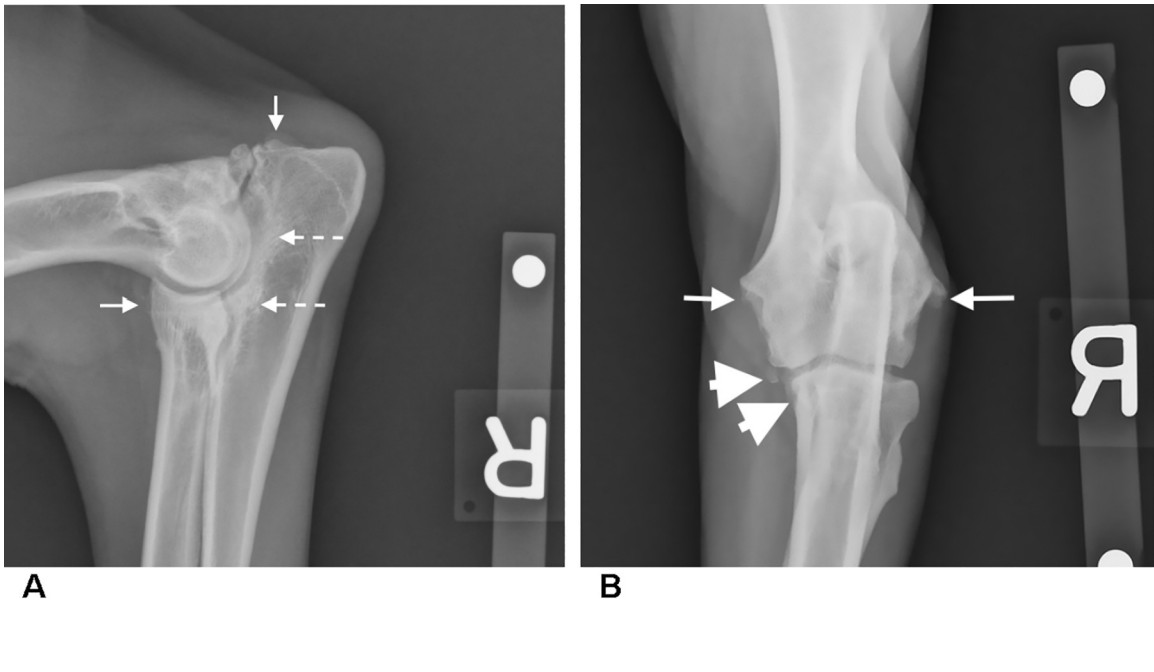

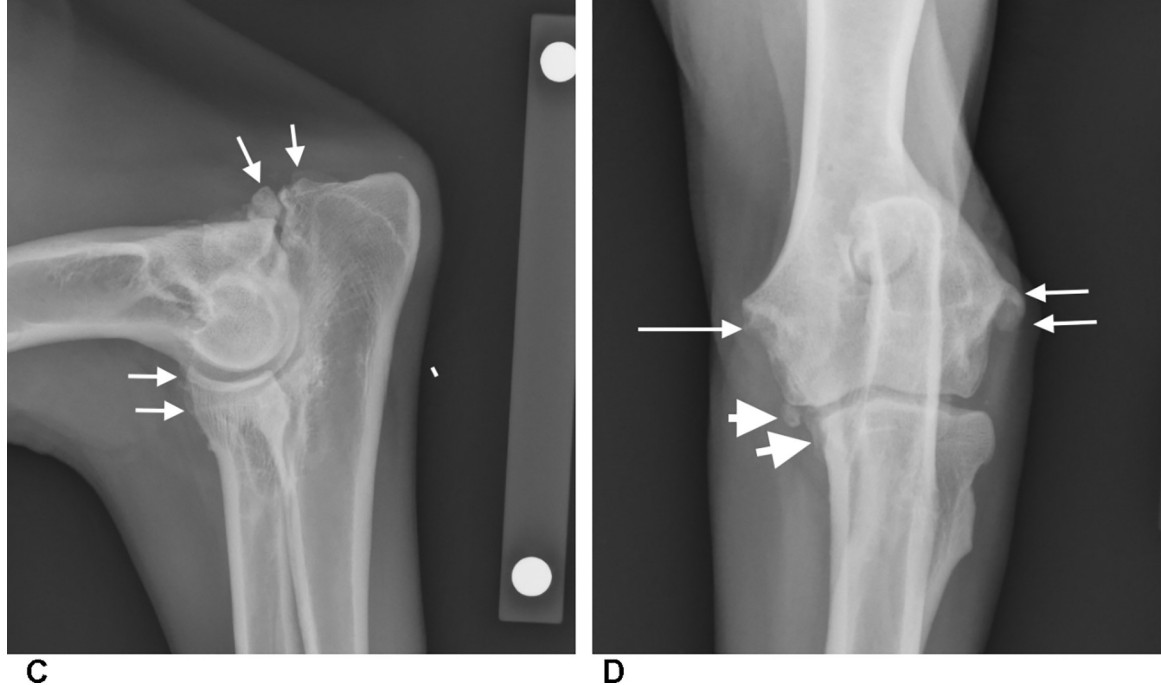

**Fig 1. Baseline and Day 365 radiographic comparison of the right elbow of dog 2-15L3.** Mediolateral (ML, view 1A) and craniocaudal (CC, view 1B) radiographs of the right elbow are shown at baseline. In the ML view, sclerosis of the trochlear notch of the ulna is present (dotted arrows). An irregular gap is present between the anconeal process and the olecranon due to an ununited anconeal process. In the CC view, moderate periarticular osteophytosis (arrows) are present at the humeral epicondyles, and anconeal process. The margins of the medial coronoid process are blunted and there is a small osseous fragment present (arrow heads). In the ML view at Day 365 (1C), there is mild progression with osteophytes at the cranial border of the radial head and epicondyles of the humerus, and anconeal process at the site of the ununited anconeal process (white arrows). In the Day 365 CC view (1D), the osseous fragment at the medial coronoid process (arrows) is slightly increased in size and the blunted process is unchanged (arrow heads). Sclerosis of the trochlear notch is static.

**Table 5. Dog owner and clinician assessments of the treatment response to tin-117m intra-articular injection in the right elbow of Dog 2-15L3.**

| Assessment | Study Interval and Test Score | | | | |
|---|---|---|---|---|---|
| | Baseline | Day 90 | Day 180 | Day 270 | Day 365 |
| **Dog owner assessments** | | | | | |
| CBPI pain severity score | 5.50 | 1.00 | 0.50 | ND | 2.25 |
| CBPI pain interference score | 1.17 | 0.17 | 0.17 | ND | 0.67 |
| **Clinician assessments** | | | | | |
| Lameness at walk[a] | 2 | 1 | 2 | ND | 2 |
| Lameness at trot[a] | 2 | 1 | 2 | ND | 2 |
| Force plate peak vertical force (Ns/kg) | 96.52 | 101.54[b] | 89.01 | ND | 70.90 |

CBPI = Canine Brief Pain Inventory; ND = not done.

[a]Scored on a 0–5 scale where 3 = severe weight-bearing lameness.

[b]Significantly improved compared to baseline, $p < 0.05$.

this population by dog owners and attending clinicians determined that a majority of dogs had a favorable response following a single IA tin-117m colloid treatment. Pain reduction scores over the 1-year duration of the study indicated that RSO with tin-117m colloid achieved a durable therapeutic effect. Dog-owner assessments of pain indicated a treatment success rate of at least 70% at all post-treatment time points. Clinician assessments showed reductions from BL in mean pain scores for 270 days following treatment, with a significant improvement on Day 90. Dog owners reported significant reductions from BL in mean PSS and PIS scores at Day 270, and a significant reduction from BL in mean PSS score at Day 365.

Although the two assessments were conducted by different groups of individuals (dog caregivers and clinical investigators) using different assessment parameters, both of the assessments focused on measurement of pain-impaired activity or movement. The statistical agreement between the two assessments at Days 90 and 270 indicates that dog owners and attending clinicians independently observed post-treatment improvement in lameness in a majority of the study population.

The study demonstrated therapeutic benefits of RSO treatment in advanced cases of DJD, namely grade 3 OA affecting a weight-bearing joint. Treatment with tin-117m may have even greater potential to modify disease progression if used in early-stage, grade 1 or 2 cases of OA before pathology has progressed to a chronic stage. Further studies of tin-117m RSO in canine populations with early-onset or pre-clinical OA would be a useful topic for future clinical research.

As with any radio-therapeutic device, safety is a paramount concern. Although reports of serious complications following RSO are rare, a European study described 19 cases of osteonecrosis in 93 human patients following RSO during a 13-year span from 1995–2007 [33]. Importantly, the RSO colloids administered to the patients were the β-emitting radionuclides yttrium-90 or rhenium-186 [18, 20]. Beta particles are higher-energy electrons with a wide tissue-penetrating range of 50–5,000 μm, making precise dosing difficult and possibly exposing non-synovial, periarticular tissue to radiation. In contrast, the novel tin-117m radionuclide emits low-energy conversion electrons with a short, well defined tissue penetration range of ~300 μm [24–26], approximating the synovial thickness. This characteristic makes precise, repeatable dosing with tin-117m possible, and reliably avoids adverse side effects affecting non-target tissue. Thus, the absence of any systemic or local treatment-related adverse effects following RSO with tin-117m was a noteworthy outcome of our study but was not unexpected.

A limitation of the study was the absence of a placebo or negative control group. This was unavoidable in the case of a radiotherapeutic device, where the dog owner would have to be advised for ethical reasons that the patient was treated with a radioisotope. Furthermore, using a placebo for a year-long study in patients refractive to previous pain treatment would be contrary to the principle of compassionate care. For purposes of this study, the BL assessment was considered to be the control benchmark. Observing a putative placebo effect was not possible due to the study design. However, it would be implausible to expect an observer placebo effect, both by dog owners and clinicians, to persist for the 1-year duration of the study. In view of these limitations, the study is appropriately characterized as observational rather than experimental in design.

The percentage of dog owners who considered treatment to be successful was somewhat greater at all time points than the success rate reported by attending clinicians. This suggests that the on-site, direct observation of functional pain impairment by clinicians was somewhat more discriminating and evidence-based than the owner's subjective assessment of their dog's level of pain. At BL, clinicians reported mean pain scores of 2.14 (walk) or 2.15 (trot), midway in the 6-point pain scale, indicating obvious weight-bearing lameness. This compared with a BL mean "worst pain" score of 7.5, at the lower margin of the first quartile of the CBPI scale, and a minimum "least pain" score of 4.5 reported by dog owners at BL, indicating a high level of persistent pain prior to treatment. Regardless, both pain assessment methods were in general agreement, with a significant statistical correlation in assessments of treatment response at the Day 90 and Day 180 time points.

Based on these outcomes, clinicians can assume that a client's CBPI assessment of a dog's condition is a valid and useful component of the patient's condition at presentation, likely to be confirmed by diagnostic imaging (all cases in the PP population had radiographically and MRI-confirmed OA), and likely to respond to RSO with tin-117m colloid. As expected in a clinical setting, the majority of the study dogs received various prescription or OTC medications for chronic treatment of joint pain during the study. Our results support RSO with tin-117m colloid as an independent OA treatment modality or as a component of a multimodal approach. In the latter case, because the activity of tin-117m colloid is confined to the synovial membrane, it has no contraindications for use as a co-therapy with agents such as NSAIDS or local or systemic corticosteroids.

## Conclusion

Study results indicated that homogeneous tin-117m ($^{117m}$Sn) colloid administered in a single IA dose provided a significant reduction in pain and lameness and improved functionality for up to a full year, with no adverse treatment related effects, in a high percentage of dogs with advanced, clinical OA of the elbow joint.

## Acknowledgments

The authors thank Dr. Sara Keeton and Marie Brajot for their assistance in compiling study data; Frank Andrews, DVM, MS, DACVIM-LAIM, Grayson Cole, DVM, DACVS, Brian Beale DVM, DACVS and Caleb Hudson, DVM, MS, DACVS for their participation in the study; and J. Lattimer, DVM, MS, DACVR-Radiology and Radiation Oncology for his guidance, especially the image analysis. Joan MacDonald of Georgetown Clinical Consulting, Atlanta, GA designed this study and Dr. Sheila Gross provided statistical assistance. Mark Dana of Scientific Communications Services, LLC prepared the manuscript.

## Author Contributions

**Investigation:** Michelle Fabiani, Lorrie Gaschen, Karanvir Singh Aulakh.

**Project administration:** John Donecker.

**Writing – original draft:** John Donecker.

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
