## [Decision Letter · Decision Letter 0]

28 Apr 2021

PONE-D-21-08558

"Treatment response in dogs with naturally occurring grade 3 elbow osteoarthritis following intra-articular injection of 117mSN (tin) colloid"

PLOS ONE

Dear Dr. Donecker,

Thank you for submitting your manuscript to PLOS ONE. After careful consideration, we feel that it has merit but does not fully meet PLOS ONE’s publication criteria as it currently stands. Therefore, we invite you to submit a revised version of the manuscript that addresses the points raised during the review process.

Hope you are well. I thank you in advance the possibility to read your work. After a careful reading of the manuscript PONE-D-21-08558, entitle “Treatment response in dogs with naturally occurring grade 3 elbow osteoarthritis following intra-articular injection of 117mSN (tin) colloid”, it is my opinion that the manuscript needs to go under major revision before being accepted to publish.

As you can check, one of the reviewers reject and the other accept with minor revision. 

The study is interesting and it focuses on osteoarthritis pain reduction in dogs using a novel therapeutic delivery intra-articular, designed in a longitudinal nature. Introduction, M&M, and Discussion sections are ok. However, there are some flaws to consider:

A brief description of the anesthetic protocol as well as the reference of the therapeutics that each patient used in the study is missing along with the manuscript. 

Also, the illustration of the patient evolution should be provided with XR films made along with the study or at least at the beginning and the end. The authors only present 2 RX films very poor in quality! 

P-value present four decimal which is not need, only two are required

Also statically issues pointed out by one of the reviewers are important to take into consideration. Maybe a revise in statistical analysis of the data might be needed to correct some of the flaws that that study presents. 

Besides these considerations, I still think that the work it is interesting and if Author’s can address some of these issues in a major revision context, the manuscript can become more fluid and correct to be able to be published. 

For this reason, is my opinion that the manuscript should go under major revision.

We look forward to receiving your revised manuscript.

Kind regards,

L.Miguel Carreira, PhD, MSc,DTO,Ps-Grd,DMD,DVM

Academic Editor

PLOS ONE

Journal Requirements:

Exubrion Therapeutics was the sole funding source for this study.  Dr. John Donecker, Chief Veterinary Officer for Exubrion Therapeutics in association with Georgetown Clinical Consulting, Atlanta, GA designed the study at the request of Exubrion Therapeutics.  Investigators at both centers adhered to the animal welfare standards as approved by the Louisiana State University Institutional Animal Care and Use Committee.  This study was funded by amendment 2 of LSU protocol number 44181(KA) and the Veterinary Specialists of Texas, PC Clinical Site Agreement dated 6/15/2017 (MF).   Owners of enrolled dogs signed consent forms indicating agreement for experimental RSO treatment and periodic reevaluations during the 1-year duration of the study.

Exubrion Therapeutics was not involved with data collection or analysis however statistical review was conducted by Dr. Sheila Gross and funded by Exubrion Therapeutics. The decision to publish and prepare this manuscript was funded by Exubrion Therapeutics.

We note that you received funding from a commercial source: Exubrion Therapeutics

3a) If there are ethical or legal restrictions on sharing a de-identified data set, please explain them in detail (e.g., data contain potentially identifying or sensitive patient information) and who has imposed them (e.g., an ethics committee). Please also provide contact information for a data access committee, ethics committee, or other institutional body to which data requests may be sent.

3b) If there are no restrictions, please upload the minimal anonymized data set necessary to replicate your study findings as either Supporting Information files or to a stable, public repository and provide us with the relevant URLs, DOIs, or accession numbers. Please see http://www.bmj.com/content/340/bmj.c181.long for guidelines on how to de-identify and prepare clinical data for publication. For a list of acceptable repositories, please see http://journals.plos.org/plosone/s/data-availability#loc-recommended-repositories.

Reviewers' comments:

Reviewer's Responses to Questions

**Comments to the Author**

1. Is the manuscript technically sound, and do the data support the conclusions?

Reviewer #1: Partly

Reviewer #2: Yes

2. Has the statistical analysis been performed appropriately and rigorously? 

Reviewer #1: No

Reviewer #2: Yes

3. Have the authors made all data underlying the findings in their manuscript fully available?

Reviewer #1: Yes

Reviewer #2: Yes

4. Is the manuscript presented in an intelligible fashion and written in standard English?

Reviewer #1: Yes

Reviewer #2: Yes

5. Review Comments to the Author

Reviewer #1: This is an uncontrolled longitudinal study of pain reduction with a secondary analysis of agreement of assessment methods. I have some concerns about the design and statistical aspects of the study. While longitudinal, the data are not analyzed longitudinally. Instead, a p-value is calculated at each time point. Presumably p<0.05 is an indication that of success. However, p-values have little meaning in the context of multiple testing, and it is not clear what is the specific hypothesis being tested to determine the specific aim of the study. The authors rightly point out that they are unable to compare to a control, so this is largely just an exploratory analysis at each time point with p-values being calculated, with no concern about type I error control. Attendant to that, there is no statistical justification of sample size or discussion of power or type I error preservation, likely because it is not clear what specifically is being tested, except using p-values as a guide to see if there is pain reduction. Similarly, McNemar's test is used to determine agreement, where agreement is based on a p-value > 0.05. Since a nonsignificant p-value does not prove the null hypothesis, it again serves only as a guide or experimental metric to indicate that there does not appear to be significant disagreement in the assessments at alpha=0.05. The paper would be have been strengthened by determining a criterion for improvement based on a hypothesis test, which could be a significant rate of change parameter from a repeated measures regression model with property controlled type I error rate. Under that hypothesis, one could then see if there is sufficient power to draw conclusions based on the sample of dogs selected.

Reviewer #2: The manuscript was good but there are somethings need more illustration

Introduction was good

Materials and Methods were good

Results were good but where is the radiographic images of affected joints before and after treatment at different intervals

Discussion was good

6. PLOS authors have the option to publish the peer review history of their article (what does this mean?). If published, this will include your full peer review and any attached files.

Reviewer #1: No

Reviewer #2: **Yes: **Mustafa Abd El Raouf

---

## [Author Response · Author response to Decision Letter 0]

16 Jun 2021

Revised manuscript cover letter: As requested in the Editorial Manager’s e-mail message of April 28, 2021, the authors are providing:

• An amended Competing Interests Statement that explicitly identifies Exubrion Therapeutics as a commercial funder of the study.

• Confirmation that we adhere to all PLOS ONE policies on data and materials sharing. All supporting references cited in the manuscript are published and available in the public domain. 

• A statement that there are no ethical or legal restrictions on data sharing.

---

## [Editor Report · Decision Letter 1]

30 Jun 2021

"Treatment response in dogs with naturally occurring grade 3 elbow osteoarthritis following intra-articular injection of 117mSn (tin) colloid"

PONE-D-21-08558R1

Dear Dr. **John Donecker **,

We’re pleased to inform you that your manuscript has been judged scientifically suitable for publication and will be formally accepted for publication once it meets all outstanding technical requirements.

Kind regards,

L.Miguel Carreira, PhD, MSc,DTO,Ps-Grd,DMD,DVM

Academic Editor

PLOS ONE

Additional Editor Comments (optional):

Dear Author,

Thank you for submmitted the revised manuscript "Treatment response in dogs with naturally occurring grade 3 elbow osteoarthritis following intra-articular injection of 117mSN (tin) colloid". After having read it and verified the introduction of the recommended changes, as well as the answer to the raised questions, it is my opinion that the article is now complete and therefore likely to be published in PLOS ONE.

Best regards
---

## [Editor Report · Acceptance letter]

9 Jul 2021

PONE-D-21-08558R1 

Treatment response in dogs with naturally occurring grade 3 elbow osteoarthritis following intra-articular injection of ^117m^Sn (tin) colloid 

Dear Dr. Donecker:

I'm pleased to inform you that your manuscript has been deemed suitable for publication in PLOS ONE. Congratulations! Your manuscript is now with our production department. 

Kind regards, 

on behalf of

Prof.Dr. L.Miguel Carreira 

Academic Editor

PLOS ONE